# Occurrence, Function, and Biosynthesis of the Natural Auxin Phenylacetic Acid (PAA) in Plants

**DOI:** 10.3390/plants12020266

**Published:** 2023-01-06

**Authors:** Veronica C. Perez, Haohao Zhao, Makou Lin, Jeongim Kim

**Affiliations:** 1Plant Molecular and Cellular Biology, University of Florida, Gainesville, FL 32611, USA; 2Horticultural Sciences Department, University of Florida, Gainesville, FL 32611, USA; 3Genetic Institute, University of Florida, Gainesville, FL 32611, USA

**Keywords:** auxins, phenylacetic acid, PAA, growth hormone

## Abstract

Auxins are a class of plant hormones playing crucial roles in a plant’s growth, development, and stress responses. Phenylacetic acid (PAA) is a phenylalanine-derived natural auxin found widely in plants. Although the auxin activity of PAA in plants was identified several decades ago, PAA homeostasis and its function remain poorly understood, whereas indole-3-acetic acid (IAA), the most potent auxin, has been used for most auxin studies. Recent studies have revealed unique features of PAA distinctive from IAA, and the enzymes and intermediates of the PAA biosynthesis pathway have been identified. Here, we summarize the occurrence and function of PAA in plants and highlight the recent progress made in PAA homeostasis, emphasizing PAA biosynthesis and crosstalk between IAA and PAA homeostasis.

## 1. Introduction

Auxins are best known as plant hormones crucial for plant growth, development, and survival. Several endogenous auxins are found in plants [1,2,3]. Among them, indole-3-acetic acid (IAA) has been commonly used for studying auxin’s function and its biosynthesis. Phenylacetic acid (PAA) is a phenylalanine-derived auxin. Although the auxin activity of PAA was demonstrated nearly a century ago and has been detected widely in plants, PAA homeostasis and its function remain poorly understood. Here, we review the occurrence and function of PAA and the recent findings on PAA homeostasis, focusing on PAA biosynthesis.

## 2. Occurrence of PAA

Auxin activity of PAA was reported in the 1930s, however, PAA was first isolated in plants in the 1960s. Acidic fractions of aqueous extract from etiolated seedlings of *Phaseolus* were shown to promote the growth of oat coleoptile sections, and further analysis identified that these acidic fractions contained PAA [4]. Subsequently, PAA was detected in various plants. Interestingly, most organs and species accumulate PAA to significantly higher levels than IAA [5,6,7]. In Arabidopsis, the PAA contents range from 200 to 3500 pmol/gFW depending on the organs, which is higher than IAA in most organs, except silique [7,8,9,10,11] (Table 1). Other dicots, such as tomato, pea, sunflower, and tobacco accumulate around 600 to 1600 pmol/gFW of PAA in their shoots, which is also several folds greater than their IAA contents [5,12]. However, the PAA content in *Tropaeolum majus* (<16 pmol/gFW) is lower than IAA (>70 pmol/gFW) [13]. Monocots and non-vascular plants accumulate 300 to 5000 pmol/gFW of PAA, similar to other dicots [5,7,11,12,14,15] (Table 1). Taken together, PAA is widely distributed in the plant kingdom, and overall, PAA accumulates more than IAA in most plants.

## 3. Biological Function of PAA

The auxin activity of PAA was identified through three classical auxin activity tests: the pea test, cylinder test, and oat bending test [17]. All three tests revealed PAA has less than 10% of IAA activity [17]. One of the most representative functions of PAA is the promotion of root growth and development [6,8,13,18,19]. PAA induced the root formation of tomato, sunflower, marigold, artichoke, buckwheat, dahlia, and tobacco when applied to the stem [20]. PAA application promoted the formation of adventitious roots of cress hypocotyls, sugar-beet seedlings, and pea epicotyls [21]. Furthermore, leaf explants of *Ajuga bracteosa* on growth media supplemented with PAA increased the frequency of root induction and biomass [19]. Arabidopsis seedlings treated with PAA increased the formation of lateral roots, although PAA showed a 10- to 20-fold lower activity than IAA [7]. However, some studies showed stronger activity of PAA than IAA. In pea seedlings, PAA induced more lateral root primordia and emerged lateral roots and longer lengths of lateral roots compared to those of IAA [6,18].

PAA impacts the aerial parts of plants as well [7,11,22,23]. Daily exogenous application of PAA on tomato plants for 2 weeks significantly increased tomato height [22]. The PAA application led to epinastic leaves of artichokes [20], stimulated the elongation of wheat coleoptile and moss gametophore [7,23], and promoted the elongation of the *Phaseolus* internode [24]. Arabidopsis plants with increased PAA production showed elongated hypocotyls and epinastic leaves, similar to those observed in high-IAA Arabidopsis plants [11]. The supplementation of PAA also induced callus formation in tobacco, sunflower, chickpea, and lentil, but the optimal concentration of PAA for tobacco callus induction was 3–4 times higher than IAA [25,26]. Similar to IAA, PAA application to leafless cotyledon retarded the petiole abscission in cotton and inhibited ethylene evolution [16].

PAA, itself, has anti-microbial and anti-fungal activities, and the exogenous application of PAA enhances tolerance to pathogen infections, whereas increased IAA in plants enhances susceptibility to pathogens [27,28,29]. For example, the exogenous application of PAA on citrus inhibits the incidence of fungus molds caused by *Penicillium digitatum* and *P. italicum* [30]. Herbivore infestation in maize, poplar, and plumeria increases PAA production [31,32,33]. The application of PAA in oilseed rape enhances the prevention of *Sclerotinia sclerotiorum,* and the in vitro treatment with PAA demonstrated adverse effects through the disruption of the cell wall and cytoplasm in mycelia [34]. However, any biological role of PAA in a plant’s defense remains unknown.

The first step of auxin action starts from auxin sensing by the auxin receptors (TIR1 and AFBs), which results in the degradation of transcription repressors Aux/IAAs [35]. Shimizu-Mitao and Kakimoto showed PAA-dependent degradation of Aux/IAA [35]. Interaction of Aux/IAAs with auxin receptors, TIR1 or AFB2, leads to the degradation of Aux/IAA [35]. They showed that PAA induced Aux/IAA degradation with lower activity than IAA [35]. Sugawara et al. showed that PAA application rescued the growth defects of IAA-deficient plant, yuc quadruple mutant (*yucQ*) [7]. In the same study, a yeast two-hybrid assay and a pull-down assay revealed that PAA promoted the interaction of auxin receptors and Aux/IAA in vitro, suggesting that the sensing and signaling modes of PAA are similar to IAA [7].

It is noteworthy, however, that PAA does not engage in polar auxin transport [7,16,36,37,38]. The labeled PAA transport assay using pea epicotyls revealed that the transport of PAA barely occurs in both the basipetal and acropetal directions [36]. PAA applied to the apical bud of intact pea plants did not move in the long-distance basipetal transport [38]. Auxin polar transport inhibitor, naphthylphthalamic acid (NPA), inhibited IAA transport, but NPA did not affect the PAA gradient patterns in cotton, pea, and maize [7,16,37]. Unlike IAA, PAA did not form concentration gradients in response to the gravitropic stimulation via active and directional transport in maize [7]. However, PAA inhibited the IAA polar transport in the internode segments and long-distance movement of the pea apical bud [38].

## 4. PAA Homeostasis

As auxins regulate a vast array of processes, changes in the content or distribution of auxins can have profound effects on plant growth and development and, in extreme cases, can lead to severe dwarfism or sterility [39,40,41,42]. Auxin homeostasis refers to the spatio-temporal distribution of auxin throughout plant tissues and organs, which governs plant growth and development. Although several processes, including auxin transport, conjugation, and degradation, influence auxin homeostasis, de novo biosynthesis directly affects the local concentration of auxins.

### 4.1. PAA Biosynthesis

The main route of IAA biosynthesis from tryptophan in plants is the YUCCA pathway [43,44,45,46,47,48,49,50,51,52] (Figure 1a). The first step of this pathway is the conversion of tryptophan to indole-3-pyruvate (IPA) by the enzymes belonging to the Tryptophan Aminotransferase of Arabidopsis (TAA) family [53,54,55,56]. Then, flavin-containing mono-oxygenases belonging to the YUCCA (YUC) family convert the IPA to IAA [42,55,57,58,59,60]. This pathway is also believed to contribute towards PAA biosynthesis (Figure 1b), as several studies have shown that the TAA and YUC enzymes can convert phenylalanine to phenylpyruvate (PPA) and PPA to PAA in vitro, respectively [7,54,59,61,62]. YUCCA overexpression increases endogenous PAA or PAA conjugate content [7]. However, the *TAA* and *YUC* mutants sometimes show little or no change in the PAA content, despite the significant alterations to the IAA content. For example, the maize YUC1-deficient mutant *de18* has an over 90% reduction in free and total IAA contents, while the PAA content is not significantly affected, and the *tar2-1* pea mutant displays a near complete loss of free IAA but no change in the PAA levels [61]. Similarly, in Arabidopsis, the *yuc1yuc2yuc6* triple and *yuc3yuc5yuc7yuc8yuc9* quintuple mutants have wild-type levels of PAA but 40–50% reductions in the IAA content [7]. However, the TAA1 deficient mutant *wei8-1* displays a 20% and 80% reduction in PAA and IAA, respectively [7]. These findings suggest that these enzymes may function in the PAA biosynthesis in plants but may not be major players in the PAA biosynthesis in some species [35].

In plants and prokaryotes, phenylalanine biosynthesis from prephenate occurs through two routes: transamination of prephenate to arogenate, which is subsequently decarboxylated and dehydrated into phenylalanine, or decarboxylation and dehydration of prephenate to form phenylpyruvate (PPA), which is converted to phenylalanine through transamination [63,64,65] (Figure 1c). Generally, the PPA pathway is more commonly found in prokaryotes, while the arogenate pathway is the major route of phenylalanine biosynthesis in plants; however, there is evidence for both pathways existing and making significant contributions in several plant and bacterial species [63,64,65]. Thus, the modification of the PPA metabolism may impact PAA biosynthesis not only directly by being converted to PAA but also indirectly by affecting phenylalanine pools in plants (Figure 1c).

Aoi et al. demonstrated that arogenate dehydratase (ADT), the enzyme that catalyzes the conversion of arogenate to phenylalanine, affects PAA biosynthesis, as the overexpression or knockout of *ADT*s result in increased or decreased PAA contents, respectively [10]. As arogenate serves as a precursor of phenylalanine, increased ADT activity results in increases in phenylalanine-derived metabolites, such as PAA. They also detected changes in the PPA levels upon the modulation of ADT activity that followed the trends measured for the PAA content (i.e., *ADT* overexpression resulted in increased levels of PAA and PPA) [10], supporting the role of PPA as a precursor of PAA.

Another PAA biosynthesis pathway is the aldoxime pathway, using phenylalanine-derived aldoxime, phenylacetaldoxime (PAOx), as an intermediate [8,10,11,15,66]. Aldoximes, such as indole-3-acetaldoxime (IAOx) and PAOx, as well as the aldoximes derived from other amino acids, are well characterized as precursors of various specialized metabolites, such as glucosinolates, cyanogenic glycosides, and camalexin as well as nitrogenous volatiles [31,33,67,68,69,70,71,72]. Several studies have shown that IAA can be made from IAOx in Brassicales and monocots [11,73,74,75] (Figure 1a). Similarly, it was shown that PAOx is a precursor of PAA in Arabidopsis and monocots [8,11,15] (Figure 1b). Arabidopsis plants overproducing PAOx increase PAA and display altered morphology, such as epinasty leaves and elongated hypocotyls, similar to those shown in plants with increased IAA [8,11,42]. Maize and sorghum plants fed with labeled PAOx produce labeled PAA [11,15]. These findings indicate a wider distribution of the aldoxime-derived auxin biosynthesis pathway throughout the plant kingdom rather than being limited to Brassicales.

The first step of the aldoxime pathway is the production of IAOx or PAOx by the cytochrome P450 mono-oxygenases of the 79 family (CYP79s) [8,11,15,32,66]. In addition to the CYP79 enzymes, flavin-containing mono-oxygenases (FMOs) in two fern species have been shown to catalyze the conversion of phenylalanine to PAOx [76]. Aldoximes contribute to auxin pools through two routes (Figure 1a,b). In Brassicales plants, both IAOx and PAOx are precursors of family-specific glucosinolates. Glucosinolates and their degradation enzyme beta-thiol-glucosidases, known as myrosinases, are stored in separate cellular and subcellular compartments [77,78,79,80]. However, upon herbivore or pathogen attack, these compartments are compromised, resulting in glucosinolate hydrolysis by myrosinases and the rapid release of toxic metabolites, such as isothiocyanates, nitriles and epithionitriles [81,82,83]. Glucosinolate turnover has also been shown to occur in vivo without tissue damage or disruption [84,85,86,87]. Nitriles, such as indole-3-acetonitrile (IAN) and benzyl cyanide, are byproducts of glucosinolate degradation. These nitriles can then be acted upon by nitrilase enzymes to generate IAA and PAA [13,88,89,90,91,92]. On the other hand, aldoximes can be converted to auxins through the aldoxime-derived auxin pathway, which is glucosinolate-independent and is present in both the Brassicales and non-Brassicales species as maize and sorghum [11,15]. A recent study demonstrated that benzyl cyanide serves as an intermediate of PAOx-derived PAA biosynthesis in maize and sorghum [15]. Application of benzyl cyanide increases PAA in maize and sorghum, and both species convert labeled benzyl cyanide to labeled PAA [15], suggesting that nitriles may be key intermediates in both routes of the aldoxime pathway (Figure 1a,b). Unlike the YUCCA pathway, the aldoxime-derived auxin pathways do not appear to be the main route of auxin biosynthesis under normal growth conditions. For example, the Arabidopsis IAOx deficient mutant, *cyp79b2 cyp79b3* (*b2b3*) double mutant, grows normally under optimal temperatures [73], and the *CYP79A2* gene encoding the PAOx production enzyme is barely expressed in the vegetative tissue of Arabidopsis ecotype Col-0 [11]. However, at high temperatures and under salt stress, the *b2b3* mutant displays a low auxin growth phenotype [73,93], suggesting that the aldoxime pathway contributes significantly towards stress-induced auxin production. Indeed, many CYP79 enzymes have their expression induced by stressors such as herbivory or by treatment with stress hormones such as jasmonic acid [31,32,68,72,94]. Given that aldoximes often serve as precursors of defense metabolites, the aldoxime pathway may play a role in modulating plant growth during the defense response.

Aside from PPA, PAOx, and benzyl cyanide, several other metabolites have been implicated in PAA biosynthesis, although where they fit within the known biosynthetic pathways is unclear. Several labeled feeding experiments have demonstrated that phenylacetaldehyde (PAAld) is derived from the phenylalanine metabolism and produced along with labeled PAA [5,95]. PAAld biosynthesis from PPA has been shown to occur in the rose through the actions of phenylpyruvate decarboxylases [96]. Additionally, in roses as well as other species, PAAld has been shown to be directly synthesized from phenylalanine by the action of aromatic aldehyde synthases or aromatic amino acid decarobylases [96,97,98,99,100]. Once PAAld is synthesized, it can potentially be acted upon by the aldehyde oxidases to generate PAA, which has been demonstrated to have activity for PAAld in maize [101]. Another metabolite, 4-phenylbutyric acid (4PBA), was recently shown to display an auxin-like effect during plant regeneration via conversion to PAA using a mechanism independent of IBR3-catalyzed oxidation [102]. Further study showing altered PAA contents upon the removal of intermediate biosynthesis enzymes may reveal the role(s) that these metabolites play in PAA biosynthesis.

### 4.2. PAA Inactivation

Another major facet of auxin homeostasis is the conversion of active auxins to inactive forms. The inactivation of auxin not only supports the formation of auxin gradients and the maintenance of auxin levels but is also necessary to prevent cytotoxic levels of auxins from accumulating in cells. IAA inactivation proceeds through two pathways: reversible IAA conjugation (to glucose, methyl, or amino acids) and irreversible IAA conjugation (to amino acids) and oxidation, with recent findings demonstrating that amino acid-conjugated IAA is oxidized and then subsequently hydrolyzed to form oxidized IAA [103,104]. As with biosynthesis, knowledge of PAA inactivation is limited compared to our understanding of IAA inactivation. Multiple studies have demonstrated that some Gretchen Hagen 3 IAA-amido synthetase (GH3) and UDP-glucuronosyltransferase (UGT) enzymes have activity towards PAA to generate PAA conjugates, such as PAA-asp, PAA-glu, and PAA-glucose, respectively [7,105,106,107,108] (Figure 1b).

### 4.3. Metabolic Interaction between IAA and PAA

More recent studies have shown a link between IAA and PAA homeostasis. The homeostasis of IAA and PAA was shown to be maintained through the modulation of auxin conjugation, with the accumulation of PAA resulting in the induction of *GH3* or *UGT* genes that preferentially act upon IAA and vice versa [8,9]. Lynch et al. showed that PPA could, in addition to its previously defined and proposed impacts of PAA biosynthesis, impact IAA biosynthesis by serving as an amino acceptor in the TAA-catalyzed conversion of tryptophan to IPA [109]. This interaction not only promotes the production of IAA but may also impact PAA biosynthesis, as an increased flux through the PPA route of phenylalanine biosynthesis was shown to decrease steady-state levels of phenylalanine [109] (Figure 1c). Perez et al. demonstrated that the accumulation of PAA results in the transcriptional downregulation of genes related to tryptophan and IAA biosynthesis in Arabidopsis [11], demonstrating a complex regulatory network for maintaining auxin homeostasis.

## 5. Conclusions

In the past decades, several biochemical and genetic studies have identified key metabolites, enzymes, and pathways that contribute towards IAA metabolism. While many questions remain regarding PAA homeostasis, recent studies have greatly expanded our understanding of how PAA is synthesized and inactivated. The role of PPA as a metabolite linking together phenylalanine, IAA, and PAA biosynthesis has been supported by genetic studies and suggests that the PPA-derived PAA biosynthesis is more complex than the corresponding IAA biosynthetic pathway. Meanwhile, the occurrence of PAOx as a PAA precursor in Brassicales and monocots suggests that this hidden pathway may be distributed widely in the plant kingdom and contribute towards PAA homeostasis wherever PAOx is produced. Additionally, the identification of PAA-amino acid and PAA-glucose conjugates within Arabidopsis has provided mechanisms for PAA inactivation, which may be shared among other species and may employ similar pathways used in IAA inactivation. Future investigation is needed to reveal other potential PAA biosynthesis and inactivation pathways, as well as the physiological roles of these pathways.

## Figures and Tables

**Figure 1 plants-12-00266-f001:**
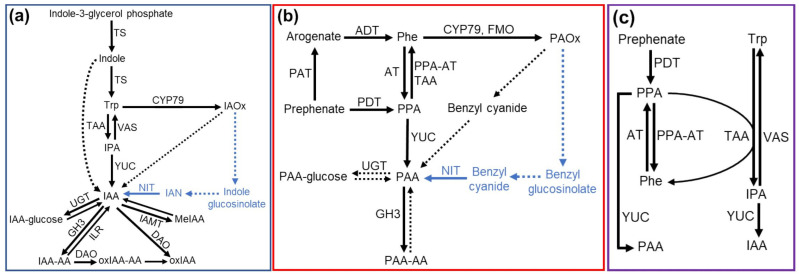
Schematic diagrams of IAA biosynthesis: (**a**) PAA biosynthesis, (**b**) and a link between IAA and PAA homeostasis in plants (**c**). The pathways only include steps and enzymes that are demonstrated via genetic evidence. Solid arrows represent single reactions catalyzed by known enzymes, and dotted arrows represent predicted single or multiple steps. Blue arrows and metabolites represent pathways and metabolites present only in Brassicales species. IAA, indole-3-acetic acid; IAA-AA, amino acid-conjugated IAA; IAN, indole-3-acetonitrile; IAOx, indole-3-acetaldoxime; IPA, indole-3-pyruvate; MeIAA, methyl-conjugated IAA; oxIAA, 2-oxindole-3-acetic acid; oxIAA-AA, amino acid-conjugated oxIAA; PAA, phenylacetic acid; PAA-AA, amino acid-conjugated PAA; PAOx, phenylacetaldoxime; Phe, phenylalanine; PPA, phenylpyruvate; Trp, tryptophan; AT, amino transferase; CYP79, cytochrome P450 mono-oxygenase of the 79 family; DAO, dioxygenase for auxin oxidation; FMO, flavin-containing mono-oxygenase; GH3, Gretchen Hagen 3 auxin-amido synthetase; ILR, IAA-Leu Resistant IAA-Amino hydrolase; IAMT, IAA carboxymethyltransferase; NIT, nitrilase; PAT, prephenate aminotransferase; PDT, prephenate dehydratase; PPA-AT, phenylpyruvate aminotransferase; TAA, tryptophan aminotransferase of Arabidopsis; TS, tryptophan synthase; UGT, UDP-glucuronosyltransferase; YUC, YUCCA family of flavin-containing mono-oxygenase; VAS; methionine aminotransferase.

**Table 1 plants-12-00266-t001:** Occurrence of PAA in plants, showing plant tissue, PAA, and IAA contents (if available).

**Species Name**	**Plant Tissue (PAA Content)**	**Plant Tissue (IAA Content)**	**Reference**
Arabidopsis (*Arabidopsis thaliana*)	Seedling (413 pmol/gFW) Dry seed (3250 pmol/gFW) Silique (800 pmol/gFW) Inflorescence (1900 pmol/gFW) Cauline leaf (400 pmol/gFW) Rosette leaf (250 pmol/gFW) Stem (200 pmol/gFW) Root (1100 pmol/gFW)	Seedling (49 pmol/gFW) Dry seed (1950 pmol/gFW) Silique (2000 pmol/gFW) Inflorescence (130 pmol/gFW) Cauline leaf (30 pmol/gFW) Rosette leaf (33 pmol/gFW) Stem (50 pmol/gFW) Root (130 pmol/gFW)	[7,8,9,10,11]
Bean (*Phaseolus vulgaris*)	Shoot	N/A	[4]
Tomato (*Lycopersicon esculentum*)	Shoot (1616 pmol/gFW)	Shoot (211 pmol/gFW)	[5,12]
Pea (*Pisum sativum*)	Shoot (632 pmol/gFW) Root (347 pmol/gFW) Cotyledon (451 pmol/gFW) Epicotyl (427 pmol/gFW)	Shoot (126 pmol/gFW) Root (115 pmol/gFW) Cotyledon (13 pmol/gFW) Epicotyl (46 pmol/gFW)	[5,6,12]
Sunflower (*Helianthus annuus*)	Shoot (1484 pmol/gFW)	Shoot (245 pmol/gFW)	[5,12]
Tobacco (*Nicotiana tabacum*)	Shoot (1234 pmol/gFW)	Shoot (228 pmol/gFW)	[5,12]
Cotton (*Gossypium hirsutum*)	Cotyledon	N/A	[16]
Nasturtium (*Tropaeolum majus*)	Root (12 pmol/gFW) Hypocotyl (14 ng/gFW) Shoot (12 pmol/gFW) Leaf stalk (13 pmol/gFW) Older leaf (11 pmol/gFW)	Root (679 pmol/gFW) Hypocotyl (166 ng/gFW) Shoot (103 pmol/gFW) Leaf stalk (74 pmol/gFW) Older leaf (86 pmol/gFW)	[13]
Sorghum (*Sorghum bicolor*)	Leaf (300 pmol/gFW)	N/A	[15]
Maize (*Zea mays*)	Shoot (903 pmol/gFW)	Shoot (143 pmol/gFW)	[5,12]
Barley (*Hordeum vulgare*)	Shoot (514 pmol/gFW) Young shoot (4353 pmol/gFW)	Shoot (63 pmol/gFW) Young shoot (30 pmol/gFW)	[5,7,12]
Oat (*Avena sativa*)	Young shoot (3860 pmol/gFW)	Young shoot (31 pmol/gFW)	[7]
Ostrich fern (*Matteuccia struthiopteris*)	Crozier (2790 pmol/gFW) Young rachis (1470 pmol/gFW) Immature pinnae (4860 pmol/gFW); Mature pinnae (2380 pmol/gFW) Fertile pinnae (2490 pmol/gFW)	Crozier (119 pmol/gFW) Young rachis (219 pmol/gFW) Immature pinnae (161 pmol/gFW) Mature pinnae (67 pmol/gFW) Fertile pinnae (70 pmol/gFW)	[14]
Moss (*Physcomitrella patens*)	1049 pmol/gFW	14 pmol/gFW	[7]
Liverwort (*Marchantia polymorpha*)	469 pmol/gFW	74 pmol/gFW	[7]

pmol/gFW is uniformly used as the unit of an approximate amount of PAA. N/A; not available in the same paper.

## Data Availability

Not applicable.

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
