# Peer review of "Occurrence, Function, and Biosynthesis of the Natural Auxin Phenylacetic Acid (PAA) in Plants"

_plants, 2023, doi:10.3390/plants12020266_

Round 1
Reviewer 1 Report
In their manuscript titled “Occurrence, Function, and Biosynthesis of the Natural Auxin Phenylacetic Acid (PAA) in Plants,” Perez, et al, review the state of knowledge of PAA, a widely occurring plant metabolite known to be able to induce auxin responses. The review briefly provides historical context for this area of study, summarizes the research into PAA levels across many plant tissues, and addresses the physiological impacts of PAA application. The manuscript then extensively reviews the metabolism of PAA, notably describing the aldoxime pathway for its synthesis.
Overall, this manuscript concisely aggregates the available research on a compound that has a growing interest among the scientific community. Although much of this has been covered in other recent reviews (notably, the 2019 review by Cook, cited as ref 63 within this manuscript), the unique focus on metabolism, and the aldoxime pathway in particular, presents a novel angle not adequately described elsewhere.
Critiques:
1) The “Occurrence” section makes comparisons of PAA levels reported in various plant tissues with IAA levels reported in various plant tissues. This comparison would be easier if the IAA levels were added to Table 1, as then the reader could directly compare PAA levels in a given tissue with IAA levels in that same tissue.
2) The “Function” section notes the apparent low auxin activity of PAA when compared to IAA. This reviewer can’t help but wonder if the auxin effects of PAA administration is truly due to direct response to PAA, or if it is indirect, resulting from PAA perturbing IAA metabolism. For example, one could envision a scenario where artificially high PAA levels are metabolically compensated for by reduced synthesis of phenylalanine and redirection of flux towards tryptophan production. Is there any evidence of for PAA directly binding to auxin-response elements? Or perhaps that PAA induces physiological responses without any discernible changes in IAA levels or distribution?
3) The authors note that PAA application confers enhanced pathogen tolerance (line 72). Is this due to direct toxicity of PAA on microbes, or from PAA inducing pathogen response pathways in the plants? This should be clarified.
4) Line 77 states that PAA is does not partake in polar auxin transport. Lines 81 to 82 then say that NPA does not inhibit such PAA transport. Please reword this section, as it is unclear how one could conclude that NPA does not inhibit transport that doesn’t happen.
Reviewer 2 Report
The authors have provided a comprehensive and generally well-written summary of the state of the field regarding the role of PAA in plant physiology, as well as the biosynthesis of this molecule. The manuscript is well-organized and the authors have done a good job of citing the relevant literature to date. The manuscript is particularly strong when discussing the biosynthesis of PAA and connections between PAA and IAA synthesis and inactivation.
Minor points:
-The manuscript could benefit from minor revisions for English grammar and usage, particularly verb tenses.
-In some cases the language used is somewhat confusion and could benefit from some clarification. For example, in line 66, the authors state that PAA accelerates cell differentiation and induces callus formation. This is a bit confusing. Are the two effects concentration- or developmental stage-dependent?
-In line 76, the authors are discussing the role of PAA in pathogen susceptibility. They state that PAA "enhances prevention" of Sclerotinia. Would it be more correct to say that the application increases resistance?
-In lines 210-211, IAA inactivation via conjugation to amino acids is discussed. The authors mention reversible conjugation, but irreversible conjugation is not discussed. It would be useful to include discussion of this aspect of IAA inactivation.
Reviewer 3 Report
General Comment:
The review covers both the historical background and the latest findings on phenylacetic acid, one of the less studied auxins to date. The review is divided into 3 main sections: Occurrence of PAA; Function of PAA; and PAA Homeostasis.
For a more comprehensive overview of the topic, I would suggest making the subtitles more descriptive and possibly dividing the "PAA homeostasis" part into several paragraphs, focusing on PAA biosynthesis, PAA inactivation, and the metabolic interaction between PAA and IAA.
I miss information on PAA signaling (Shimizu-Mitao, Y. and Kakimoto, T. 2014, Sugawara 2015 and others). I would also include all the work by David Morris (and others) on PAA transport in a separate paragraph.
Major comments:
l. 35-36 ...PAA contents range from 200 to 3500 pmol/gFW.....which is up to 14,8 fold higher than... It seems to me to be unreasonable accuracy in terms of doses that vary within an order of magnitude.
Table 1> To show the range of PAA as well as IAA would be beneficial (updated Cook 2019 Table1 in fact
l. 46 the title "Function of PAA" might be more descriptive (suggestion>Biological Function of PAA in higher plants and non-plant organisms...)
Fig.1a ILR is missing. Direct oxidation of IAA to oxIAA cant be omited (3x PNAS 2016 Stepanova and the others...) despite the recent trends in the presentation of IAA metabolic schema.
l. 132 and 137 Figure 1c is meant, I guess.
l. 221 and 222 please specify "one auxin" and "the other auxin"
Minor comments:
Citation 10 is the same as 16
Citation 11 is the same as 22
Citations 18 and 22 are not complete.
